# Developmental Pathway Choices of Young People Presenting to a Gender Service with Gender Distress: A Prospective Follow-Up Study

**DOI:** 10.3390/children10020314

**Published:** 2023-02-07

**Authors:** Joseph Elkadi, Catherine Chudleigh, Ann M. Maguire, Geoffrey R. Ambler, Stephen Scher, Kasia Kozlowska

**Affiliations:** 1Department of Psychological Medicine, The Children’s Hospital at Westmead, Westmead, NSW 2145, Australia; 2The Children’s Hospital at Westmead Clinical School, Faculty of Medicine and Health, University of Sydney, Westmead, NSW 2145, Australia; 3Department of Endocrinology, The Children’s Hospital at Westmead, Westmead, NSW 2145, Australia; 4McLean Hospital, Department of Psychiatry, Harvard Medical School, Boston, MA 02115, USA; 5Department of Psychiatry, Faculty of Medicine and Health, University of Sydney, Camperdown, NSW 2050, Australia; 6Brain Dynamics Centre, Westmead Institute of Medical Research, Faculty of Medicine and Health, University of Sydney, Westmead, NSW 2145, Australia

**Keywords:** gender dysphoria, transgender, persistence, desistance, holistic (biopsychosocial) practice, mental health, assessment, treatment, clinical practice, outcomes

## Abstract

This prospective case-cohort study examines the developmental pathway choices of 79 young people (13.25–23.75 years old; 33 biological males and 46 biological females) referred to a tertiary care hospital’s Department of Psychological Medicine (December 2013–November 2018, at ages 8.42–15.92 years) for diagnostic assessment for gender dysphoria (GD) and for potential gender-affirming medical interventions. All of the young people had attended a screening medical assessment (including puberty staging) by paediatricians. The Psychological Medicine assessment (individual and family) yielded a formal DSM-5 diagnosis of GD in 66 of the young people. Of the 13 not meeting DSM-5 criteria, two obtained a GD diagnosis at a later time. This yielded 68 young people (68/79; 86.1%) with formal diagnoses of GD who were potentially eligible for gender-affirming medical interventions and 11 young people (11/79; 13.9%) who were not. Follow-up took place between November 2022 and January 2023. Within the GD subgroup (n = 68) (with two lost to follow-up), six had desisted (desistance rate of 9.1%; 6/66), and 60 had persisted on a GD (transgender) pathway (persistence rate of 90.9%; 60/66). Within the cohort as a whole (with two lost to follow-up), the overall persistence rate was 77.9% (60/77), and overall desistance rate for gender-related distress was 22.1% (17/77). Ongoing mental health concerns were reported by 44/50 (88.0%), and educational/occupational outcomes varied widely. The study highlights the importance of careful screening, comprehensive biopsychosocial (including family) assessment, and holistic therapeutic support. Even in highly screened samples of children and adolescents seeking a GD diagnosis and gender-affirming medical care, outcome pathways follow a diverse range of possibilities.

Over the last decade, across Australia, Europe, and the United States, the rates of young children (including adolescents) presenting with distress concerning sex assigned at birth (“gender-related distress”) have continued to increase [1,2,3,4,5,6,7,8]. In a subset of these children, this distress is coupled with impairment in social, occupational, or other important areas of functioning, and the child and family are likely to seek help within the health care system. If the distress is substantial, the child may be eligible for a formal diagnosis of Gender Dysphoria (GD) [9] and a range of treatment options yielding different developmental pathways and choices. In the current article we report on the developmental pathway choices—and clinical outcomes—of 79 young people who first presented to our hospital’s gender service 4–8 years previously (December 2013–November 2018) [10]. We also discuss the changes in thinking, clinical recommendations, and clinical practice that have taken place, across the world, in the field of gender dysphoria over the last decade. For a definition of the terminology used in this article see Text Box 1. 

Box 1Definition of Terms.*Biological sex:* refers to the pattern of findings on chromosomal testing. An XY chromosomal pattern refers to male sex (♂), and an XX chromosomal pattern refers to female sex (♀). All participants in the study had chromosomal testing as part of their medical workups.*Cisgender:* refers to a gender identity that is aligned (congruent) with biological sex. *Desistance in the cohort as a whole:* In the cohort as a whole, *desistance* refers to the resolution/disappearance of the gender-related distress that was the foundation for the young person to present to the service. *Desistance in the Gender Dysphoria subgroup:* In the subgroup with a formal diagnosis of Gender Dysphoria (DSM-5), *desistance* refers to discontinuation of the journey to transition to the other gender (transgender pathway). In the gender dysphoria subgroup, the act of desisting from the transgender pathway included cessation of social transition, puberty blockers, or cross-sex hormones or a combination of these elements.*Gender:* refers to each participant’s subjective experience of identity along the gender spectrum.*Gender Dysphoria (GD):* refers to a feeling of distress (dysphoria) that meets diagnostic criteria for gender dysphoria as per DSM-5 [9].*Gender-related distress:* refers to a feeling of distress (dysphoria) pertaining to gender that may or may not meet DSM-5 criteria for gender dysphoria.*Persistence:* refers to continuation of the journey to transition to the other gender (transgender pathway). In the current cohort, persistence could include social transition, treatment with puberty blockers, treatment with cross-sex hormones, gender-affirming surgery, or any one element or of a combination of elements.*Transgender:* refers to a gender identity that is not aligned with biological sex but is instead aligned with the other sex. In the case of the participants from the current cohort, experiencing the self as transgender was the foundation for the subjective experience of gender dysphoria that met the DSM-5 criteria for gender dysphoria.

The Gender Service at the Sydney Children’s Hospitals Network is a multidisciplinary service located in a tertiary care children’s hospital in New South Wales. The service was established in December 2013 in response to the increased referrals—including court-mandated orders—to our hospital’s Endocrinology Department for children with gender dysphoria seeking treatment with puberty-suppressing medications.

At that time, two sets of published guidelines were available to guide clinical practice [11,12]. The guidelines suggested a model of treatment for younger, prepubertal children that involved the following: provision of information, psychological support, and parental or family counselling. In children for whom the gender dysphoria persisted and who were distressed by the development of secondary sex characteristics with the onset of puberty, puberty was suppressed with gonadotropin-releasing hormone agonists (GnRHa, stage 1 of a medical intervention), while children 16 years and over were offered *medical* gender-affirming treatment (with cross-sex hormones) (stage 2 of a medical intervention) and a subsequent option for *surgical* gender-affirming treatment (stage 3 of a medical intervention). This treatment approach was originally developed in the Netherlands in the late 1980′s to mid-1990s [13,14] and over time came to be widely adopted as global standard practice for the treatment of children and adolescents diagnosed with gender dysphoria [15,16]. 

Despite the existence of guidelines, in 2013 our clinical team found that the evidence base for all aspects of treatment was sparse, especially relating to long term outcomes. In this context, we discussed the situation with our hospital executive. Based on these discussions we structured the Gender Service to provide careful screening and holistic care using a multidisciplinary team approach [17]. The multidisciplinary approach involved the following steps.

*Step 1*. Initial phone-screening triage was undertaken by a clinical psychologist from Psychological Medicine and later by a nurse consultant from Adolescent Medicine. Only children who potentially met the Diagnostic and Statistical Manual of Mental Disorders (DSM)–5 diagnostic criteria for GD and who met the hospital intake criteria—cut-off at the sixteenth birthday—were screened in. 

*Step 2*. A first-level screening assessment (including a psychosocial assessment and puberty staging) was undertaken by a paediatrician from Adolescent Medicine. 

*Step 3*. A comprehensive psychiatric biopsychosocial assessment (individual and family) with clinicians—psychiatrist, psychologist, and registrar—from Psychological Medicine was conducted. Because the Gender Service was unfunded and did not have resources—in terms of staff—to provide psychotherapy or family therapy over time, a prerequisite for referral to Psychological Medicine was current engagement with a therapist (a psychologist, psychiatrist, or other qualified therapist). The idea behind this criterion—regardless of the outcome of the Psychological Medicine assessment—was that the child and family would require ongoing psychosocial support to work though the issues that had contributed to the child’s distress and presentation to the Gender Service.

*Step 4*. For the subgroup of children who met formal diagnostic criteria for gender dysphoria and who were actively seeking intervention along the medical pathway, step 4 involved provision of a referral to a paediatric endocrinologist for menstrual management or for consideration of puberty suppression with gonadotropin-releasing hormone agonists (GnRHa). For children and families who chose the option of puberty suppression, long-acting goserelin acetate (Zoladex) injections were given every 10 weeks. The physical effects of GnRH agonists have long been thought to be fully reversible: when ceased, puberty recommences. Recent data suggest a negative impact of long-term puberty suppression on bone mineral density (p. 1) [18,19]. Far less is known, however, about the effects of puberty suppression on cognitive, psychosocial and psychosexual development and mental health functioning [19,20,21]. 

*Step 5*. Young people meeting formal diagnostic criteria for GD, nearing the age of 16, and wanting treatment with cross-sex hormones were transitioned to adult services for further assessment regarding treatment with cross-sex hormones. Because many of the actions of cross-sex hormones are irreversible, and because a variety of adolescent-related and adult-related issues, including sexual orientation, a potential for difficulties with future sexual function, and fertility counselling and preservation, were now coming into play, it was considered of utmost importance that a new assessment be conducted by the clinician(s) who would follow the young person into adulthood.

Alongside the clinical service, a research program was established from the outset. The goal of the research program was to generate data that would provide an evidence base for guiding clinical interventions. All families presenting for assessment to the Gender Service were offered an opportunity to participate in research. Analysis of the data from the first five years of the clinical service highlighted the complexity of the clinical presentations and the many clinical challenges faced by the multidisciplinary team [10,22]. Key themes included the following: high rates of comorbid mental health concerns, complex family issues, clinician concerns pertaining to consent, and clinician concerns about the paucity of medical information about the long-term outcomes (physical and psychological and cognitive) of GnRHa and cross-sex hormones. Alongside our international colleagues [4,23,24,25,26,27,28], the founding multidisciplinary team also became aware of the increase of presentations of what was termed *late-onset*, *rapid-onset*, or *adolescent-onset* GD. This group of adolescents, predominantly female, had no prior history of gender distress during early development and presented with sudden-onset gender-related distress. The absence of prior history raised questions that this particular group of adolescents were being drawn to the construct of gender dysphoria because of some evolving social process. In particular, we wondered whether gender dysphoria provided an uncomplicated framework for understanding (and also for resolving) the inner distress that had arisen in the context of adverse childhood experiences and the challenges and existential distress associated with adolescence, especially in this turbulent, uncertain social and political period.

## 1. National Practices and Policies

In a parallel process, questions about the early guidelines—and the management of dysphoria regarding sex assigned at birth in children and adolescents—were being raised by governments and medical boards across the globe. 

### 1.1. Finland

In Finland, responding to the increase in adolescent-onset gender-related distress [4,29], the high rates of comorbid mental health concerns [4], and outcome studies suggesting that “medical gender reassignment is not enough to improve functioning and relieve psychiatric comorbidities among adolescents with gender dysphoria” (p. 213) [30], the Council for Choices in Health Care in Finland (COHERE Finland) met over a 12-month period to develop a recommendation pertaining to the treatment of children with gender-related distress (released June 2020) [31]. The Finnish recommendations highlighted the importance of psychosocial support provided in school and primary health care settings—including consultation with a child and adolescent psychiatrist as indicated, coupled with any necessary psychiatric treatment and psychotherapy. Assessment for medical interventions was restricted to the specialised, multidisciplinary, tertiary-level units of Helsinki University Central Hospital and Tampere University Hospital. The recommendation endorsed the idea that “surgical treatments are not part of the treatment methods for dysphoria caused by gender-related conflicts in minors” (p. 2) [31]. Following publication of these guidelines, clinicians working in the two above clinics have required the following conditions to be met before undertaking the full diagnostic assessment for possible GD and eligibility for medical interventions: completion of psychosocial intervention (to be described in referral) to support identity exploration; treatment to remission of any severe mental disorders; and the young person’s entry into at least the early stages of puberty (personal communication, Rittakerttu Kaltiala).

### 1.2. Sweden

In Sweden the National Board of Health and Welfare (NBHW) was commissioned in response to similar concerns and to develop updated guidelines. Compared with the previous guidelines (2015), the updated guidelines—first published in March 2021, and updated again February 2022 and December 2022—adopted a more constrained approach for adolescents with gender-related distress. “The NBHW deems that the risks of puberty suppressing treatment with GnRH-analogues and gender-affirming hormonal treatment currently outweigh the possible benefits, and that the treatments should be offered only in exceptional cases. This judgement is based mainly on three factors: the continued lack of reliable scientific evidence concerning the efficacy and the safety of both treatments, the new knowledge that detransition occurs among young adults, and the uncertainty that follows from the yet unexplained increase in the number of care seekers, an increase particularly large among adolescents registered as females at birth” (p. 3) [32]. Exceptional cases were those that met three key criteria: onset of gender-related distress in childhood; persistence over time; and high levels of distress with the commencement of puberty. The guidelines also emphasized the importance of “complex multidisciplinary assessments” within specialised settings—three have been designated—and note that medical treatment should take place within the framework of research, thereby ensuring the generation of knowledge [32].

### 1.3. United Kingdom

In the United Kingdom, also in response to similar concerns, the National Health Service (NHS) commissioned an independent review—the Cass Review led by Dr Hilary Cass, past president of the Royal College of Paediatrics and Child Health—of gender identity services for children and young people in September 2020 [33,34]. The review was established in response to the following: “the significant and sharp rise in referrals”; “the marked changes in the types of patients being referred which are not well understood”; “scarce and inconclusive evidence to support clinical decision making”, and “long waiting times for initial assessment and significant external scrutiny and challenge surrounding the clinical approach and operational capacity at [Gender Identity Development Service]” [33]. The above problems had contributed to the NHS service being unable to meet the scale of rising demand [33]. An interim report from the Cass Review (released in February 2022) yielded the following recommendations [33,34]:−Closure of the Tavistock service (a centralised NHS service) [35,36] and the opening of “Regional centres [that] should be led by experienced providers of tertiary paediatric care to ensure a focus on child health and development, with strong links to mental health services. These will generally be specialist children’s hospitals”.−Services “should have established academic and education functions to ensure that ongoing research and training is embedded within the service delivery model”.−Services “should have an appropriate multi-professional workforce to enable them to provide an integrated model of care that manages the holistic needs of this population”.−“Staff should maintain a broad clinical perspective to embed the care of children and young people with gender uncertainty within a broader child and adolescent health context”. Along these lines, the Cass Review noted that “We also welcome the recognition that this is a heterogenous group and that not all children and young people will want or require a medical pathway, and that the service needs to include the appropriate skill mix to support both those individuals who do require medical intervention and those who do not” (p. 2) [34].

### 1.4. Australia

In Australia the situation is evolving. In the state of Victoria, Telfer and colleagues—from Royal Children’s Hospital Melbourne—published a position statement entitled “Australian Standards of Care and Treatment Guidelines for Transgender and Gender Diverse Children and Adolescents” in the *Medical Journal of Australia* [37,38]. The title is actually misleading. In Australia there are no official or authorized government-commissioned standards for assessing or treating gender dysphoria. The document by Telfer and colleagues supports the gender affirmative model [15,16,39,40], in which “decision making should be driven by the child or adolescent wherever possible; this applies to options regarding not only medical intervention but also social transition” (p. 133). In the state of New South Wales, the National Association of Practicing Psychiatrists (NAPP), developed guidelines that suggest an alternative model of care. The NAPP guidelines recommend that “individualised psychosocial interventions (e.g., psychoeducation, individual therapy, school-home liaison, family therapy) should be first-line treatments for young people with gender dysphoria/incongruence” and that these treatments “should be undertaken before experimental puberty-blocking drugs and other medical interventions (e.g., cross-sex hormones, sex reassignment surgery) are considered”. The guidelines highlight how this approach is “consistent with established principles of comprehensive, systemic youth health care”. The two documents highlight the difference in opinions that are part of the ongoing medical discourse in Australia.

More recently, the NSW Ministry of Health commissioned Urbis—a private consulting firm that has conducted wide-ranging social, political, economic, health care, and educational projects—to create a “Summary of Evidence based on an extensive needs assessment process conducted during 2019–20” (back cover) [41] to yield a NSW health plan for lesbian, gay, bisexual, trans and gender diverse, intersex, queer, and questioning (LGBTIQ+), including children and adolescents with GD [42]. The voices, experiences, and needs of individuals who desisted or detransitioned are not included in the documents. In relation to individuals with GD—including children and adolescents—the Summary of Evidence document takes a non-conservative direction that is at variance with the Finnish and Swedish guidelines, and the Cass Report in the United Kingdom:−Improved access to gender-affirming treatments and care is therefore a key priority. Almost three-quarters of transgender and gender diverse respondents to our LGBTQ community survey indicated difficulties accessing such services (71%). Barriers to access include the limited number of services in NSW, the high costs of some treatment options such as puberty blockers and surgeries, and the requirement for a diagnosis of ‘gender dysphoria’ by a psychiatrist to access hormone replacement therapy (HRT) (p. 13) [41].−The strategy should enable a pathway of care for people seeking to affirm their gender. The pathway of care should focus on depathologising and reducing barriers to accessing gender-affirming treatments and care. The pathway of care should centre on the expertise, informed consent, rights and lived experience of transgender and gender diverse adults, adolescents and children (p. 20) [41].−[Depathologising] refers to moving away from classifying transgender people as having a mental health condition such as ‘gender dysphoria’ and from the requirement of a diagnosis of gender dysphoria before access to gender-affirming treatments and care is permitted (footnote bottom of page 20).

Of note, despite their use in practice, no drugs have been approved by Australia’s Therapeutic Goods Administration (TGA) or subsidised by the Pharmaceutical Benefits Scheme (PBS) for the treatment of gender dysphoria. All such prescriptions are therefore “off-label”.

### 1.5. United States

In the United States, the situation is complex. Although the federal Food and Drug Administration has approved no drugs for use in GD, GnRH-analogues and gender-affirming hormones are widely used anyway. The one other potential source of federal control regards the availability (or not) of federal funding for particular gender-related services. Otherwise, long-established practice is that such matters are left to professional organizations (either nation- or state-wide) and, indeed, to individual physicians in the absence of clear, binding guidelines.

At present, policies vary widely among the 50 states. The two most commonly cited policies/guidelines regarding GD (both supportive of gender-affirming interventions, and both nonbinding) are the World Professional Association for Transgender Health’s *Standards of Care for the Health of Transgender and Gender Diverse People*, now published in Version 8 [43], and the Endocrine Society’s 2017 Clinical Practice Guideline [16], reaffirmed in the society’s 2020 position statement *Transgender Health* [44]. Various states have enacted legislation restricting the use of puberty blockers and hormones in minors; for example, a recently enacted Alabama statute criminalizes distribution of puberty blockers and hormones to anyone under the age of 19. But all of these statutes have been challenged in court. None has actually gone into effect.

The state of Florida’s Department of Health has taken a different approach: to establish state-wide practice standards by directly regulating the medical profession. After commissioning an evidence review that found the evidence supporting hormonal and surgical interventions to be of low quality [45], the department classified such interventions to be experimental [46] (a finding now being challenged in court), with the consequence that funds from Medicaid, a joint federal–state program for economically disadvantaged children and adults, cannot be used to pay for the interventions. More interestingly, the Florida boards of medicine (one for medical doctors and one for osteopathic-trained doctors) both barred the use of hormones and surgery under the age of 18, though the osteopathic board allowed for hormonal and surgical research under that age. Unlike the guidelines/policies of the World Professional Association for Transgender Health and the Endocrine Society, these Florida standards are legally binding (at least within the state).

### 1.6. Summary

An important theme from the Finnish, Swedish, and UK guidelines and recommendations is the need for research [31,32,33,34], including, in particular, the collection of short- and long-term outcome measures that examine the impact of gender-affirming medical treatment on the felt sense of gender dysphoria, mental health, quality of life, and physical wellbeing, including the treatment’s risks regarding bone health, metabolic outcomes, sexual function, fertility, cognitive and emotional development, and other health outcomes.

A key problem at present is that data from outcome studies are sparse, inconsistent, and low in terms of evidence-based gradings—especially with regard to children and adolescents [47,48,49,50,51,52,53,54]. Another major concern is the lack of longitudinal studies that document the developmental trajectories and physical and mental health outcomes of all participants, including those that continue along a GD pathway and those that do not. Concerns pertaining to consent have also been raised [10,52,55,56,57].

The emerging voices of detransitioners have identified important issues [57,58,59,60,61,62,63,64,65,66]. Some have reported that they had come to believe that gender-affirming medical treatment would alleviate their feelings of dysphoria, but it had not. Some have highlighted the potential for adverse outcomes, particularly in relation to interventions whose effects cannot be reversed (i.e., cross-sex hormones and gender-affirming surgery). Some have reported that, in hindsight, because of their age or mental health concerns, they were not fit to give consent at the time that it was required. Some raise regrets about making decisions about their sexuality before that sexuality—and their understanding of that sexuality—was explored and clarified. Finally, some think—in retrospect—that they were misguided in focusing exclusively on their gender dysphoria: that they should also have considered and addressed some of the concurrent adverse childhood experiences and issues pertaining to peer relationships and emerging sexuality that were contributing to their subjective distress and loss of wellbeing. In this scenario the professionals’ affirmation of the gender dysphoria was seen as simplistic and superficial, reflecting a failure to take a more in-depth approach and to examine what was going on underneath.

Amid the ongoing controversies involving interventions for GD [52,67,68,69,70,71], the Gender Service at the Sydney Children’s Hospitals Network has tried to follow a cautious, carefully considered, individually tailored approach. As an initial step, the young people who presented to our service from December 2013 to November 2018—typically in an effort to obtain gender-affirming medical interventions—were assessed for GD and then, on that basis, became eligible or not to proceed with puberty suppression with GnRHa through our Gender Service. During this time period, all young people were required to have a therapist—from their local communities—who would support the ongoing process of identity exploration and therapeutic intervention regarding comorbid concerns. Our Gender Service did not have the capacity to provide ongoing therapeutic work. Young people who subsequently chose to pursue the option of cross-sex hormones were referred for further assessment and potential medical intervention to adult services. These young people have been subsequently making choices about the direction of their lives, thereby establishing their own developmental pathways. In the current study we report on the developmental pathways, educational/occupational function, and mental health of this cohort of young people.

## 2. Methods

From December 2013 to November 2018, when children and their families presented to the Gender Service of the Sydney Children’s Hospital Network for assessment and potential treatment, they were also given the opportunity to participate in a research project documenting clinical presentations, clinical pathways, and outcomes using a naturalistic approach. The naturalistic approach is a qualitative research method where outcomes of the research participants in the real-world setting are documented. In the current instance the real-world setting was the young people’s interaction with the medical system—within the current legal framework [72]—to obtain gender-affirming medical care. The Sydney Children’s Hospital Network Ethics Committee approved the study. Participants and their legal guardians provided written informed consent. 

We undertook follow-up of this clinical cohort through November 2022–January 2023—a period of 4–9 years, depending upon when they initially presented. We used several methods to collect the follow-up data:−For all young people who had exited the Gender Service, final follow-up telephone calls with them and their families were attempted between November 2022 and January 2023. No calls were made to the young people and families who had previously requested no further follow-up calls (n = 3). The interviewer (JE) used a script to guide the questions asked during the telephone interview (see Text Box 2).−For young people who had exited the service and who could not be contacted by telephone during the November 2022–January 2023 period (see above), information was collated from past clinic letters, from letters sent by the clinicians within the adult health system to whom the young person’s care had been transitioned, and from previous follow-up phone calls up to the middle of 2021.−For young people who were still engaged in the Gender Service—that is, they had ongoing face-to-face visits in the clinic—information from recent clinic letters was used. Telephone follow-up was undertaken to clarify any missing information. 

Box 2Script template used to collect follow-up information from participants.
*Hello, my name is Dr JE from the Children’s Hospital at Westmead. Am I speaking to [patient/parent]?*

*I work with psychiatrist Dr KK in Psychological Medicine and was calling in regard to follow-up for the gender study you were enrolled in some time ago. Is it Ok if I ask a few questions?*

**Question 1—Asked only if this information was not known from patient notes or previous follow-up calls***Did you ever receive stage 1 therapy, commonly known as puberty blockers?*If yes: What age were you when they were started?**Question 2—Asked only if this information was not known from patient notes or previous follow-up calls***Did you ever receive stage 2 therapy, commonly known as cross-sex hormones or gender reaffirming hormones?*If yes: What age were you when they were started? Are you currently still taking them. If not, when were they ceased?**Question 3***Do you have any current medical or mental health conditions?*If yes: What conditions? Are they being treated?**Question 4***Are you currently working or studying? *If yes: What type of employment/study?**Question 5***Have you undergone any gender-related surgery, or are you considering surgery in the future? *If yes: What type of surgery? Age surgery occurred?

### Data Analysis

Qualitative analyses were used to report findings.

## 3. Results

### 3.1. Demographics

The final sample comprised 79 young people, now aged 13.25–23.75 years (mean = 19.00; SD = 2.50; median = 19.08), who had presented to the Gender Service (December 2013 to November 2018) with gender-related distress (see Figure 1) [10]. Thirty-three (41.8%) young people were biological males, and 46 (58.2%) were biological females (confirmed on chromosomal testing). The young people and their families had come from all parts of the state of New South Wales. On initial presentation to the Gender Service, the majority of young people (n = 61; 77.2%) had reported that they were attending the service because they were seeking a referral to Endocrinology for medical intervention—most commonly, the prescription of puberty-blocking medications. The presenting clinical characteristics of this cohort have been described in detail in a previous publication [10].

### 3.2. Information Sources Pertaining to Outcomes

Current information from phone interviews (n = 34 conducted in November 2022–January 2023); current clinic letters (n = 10; young people still attending the service), or both (n = 2); and recent correspondence from the young person’s current clinicians outside of our institution (n = 4)—was available for 50 (50/79; 63.3%) young people. Three young people (and their families) who had requested no further follow-up were not contacted. For the rest of the sample, less current information was available from past clinic reports, past letters from clinicians to whom the young person had been transitioned/referred, and previous follow-up phone follow-up calls (up to mid-2021). Two young people had been lost to follow-up.

### 3.3. The Diagnostic Assessment Process

For the majority of young people (71/79; 89.9%), the assessment process in the Gender Service included telephone triage, a medical screening assessment by a paediatrician (a 2–3-h assessment that included psychosocial history and puberty staging), and a full-day assessment with the Psychological Medicine team. The latter included a family assessment, an individual assessment, a parent session, feedback with the young person and family, and subsequent telephone contact with the child’s individual therapist, school counsellor, and any other relevant parties. For a subset (8/79; 10.1%), the above process was insufficient, and the gender-assessment process needed to be extended with additional sessions (2–10; mean = 3.75; SD = 2.66; median = 3.00). In this subgroup, assessment was more difficult because of a history of maltreatment or the presence of complex comorbid mental health concerns, raising the questions, respectively, of whether dislike of secondary sexual characteristics could potentially have been secondary to trauma or whether serious mental health problems could be affecting the perception of gender or the experience of emerging sexuality. Importantly, when we identified trauma or serious acute mental health concerns as potential issues, information exchange with the young person’s individual therapist—sometimes over an extended period—was crucial in helping the Psychological Medicine team achieve diagnostic clarity.

Of the eight young people whose assessments for GD required additional sessions, two—one who met formal diagnostic criteria for GD and one who did not—also needed further mental health assessments and required ongoing psychiatric treatment within the hospital (though not within the Gender Service). During this period of psychiatric treatment, the one with the formal GD diagnosis had two mental health admissions for suicidal ideation with high risk, combined with ongoing care through their local, community-based mental health service. The other young person (who did not meet GD criteria) had three subsequent psychiatric appointments at the hospital for assessment and treatment of major depression, generalised anxiety, restrictive eating in the context of stress, functional somatic symptoms, and post-traumatic stress disorder, followed by nine additional therapy sessions. After the hospital-based treatment of these two patients was completed, care was handed back to their community-based therapists.

### 3.4. Developmental Pathway Choices of Study Participants

In the end, the Psychological Medicine assessment yielded 66 young people (66/79; 83.5%) who met DSM-5 criteria for GD (GD group) and 13 young people (13/79; 16.5%) who did not (non-GD group). Two young persons from the latter group subsequently obtained a GD diagnosis from an adult service at 17 and 18 years of age respectively. This yielded 68 young people who were eligible to consider medical interventions and 11 who were not. For a visual representation of the choices pertaining to medical intervention made by these 79 young people, see Figure 1.

### 3.5. Treatment with Gonadotropin-Releasing Hormone Analogues (Puberty Blockers)

Through our Gender Service, 49 young people had commenced on treatment with puberty blockers, Stage 1 of the gender-affirmative medical pathway (see Figure 1). Age of commencement varied widely (9.42–15.33 years; mean = 13.26; SD = 1.49; median = 13.50), as did the duration for which the children were treated (8 months–5.75 years; mean = 2.72 years; SD = 1.42; median = 2.50 years). One child who had been started on puberty blockers for precocious puberty at 9.2 years of age—and had subsequently been maintained on them for GD (a diagnosis made later) over a period of 7.58 years—was not included in the above analyses. 

Endocrine reviews documented side effects in 23 of the 49 young people (46.9%): low bone density (7/49; 14.3%); hot flushes (8/49; 16.3%); weight gain (5/49; 10.2%); anxiety or discomfort with the injection (2/49; 4.1%); and bruising around the injection site (1/49; 2.0%). The bone density findings are reported in more detail in Table 1. Of the seven young people with low bone density, four had had low bone density prior to starting puberty blockers (further worsened by puberty blockers in all four cases) and three had normal bone density prior to starting puberty blockers and low bone density as a side effect of puberty blockers.

### 3.6. Treatment with Cross-Sex Hormones

In New South Wales, prior to 2017—the fourth year of the study—Australian law allowed prescription of cross-sex hormones for adolescents only under the umbrella of court orders. In 2017, a change in Australian laws allowed prescription of cross-sex hormones to children ≥16 years who were assessed by clinicians to be competent to provide informed consent—or if not deemed competent, to have the parent or legal guardian provide informed consent. 

Fifty-one young people had commenced on treatment with gender-affirming cross-sex hormones (see Figure 1) outside our institution. Age of commencement varied widely (13.75–19.00 years; mean = 16.15; SD = 1.05; median = 16.00). Of this group, 20 of 51 (39.2%) commenced cross-sex hormones before the age of 16 years. Because access to cross-sex hormones is restricted in New South Wales to ≥16 years of age (see above), these data suggest that this subgroup of young people and their families accessed cross-sex hormones from unregulated sources or unregulated providers. Because cross-sex hormones were not prescribed at our service, no endocrinology data about any side effects are available.

### 3.7. Gender-Affirming Surgery

At the current follow-up point, six young people reported that they had undertaken gender-affirming surgery: double mastectomy (n = 4), double mastectomy plus hysterectomy (n = 1), and breast implantation plus penile skin inversion vaginoplasty (n = 1). Three young people reported they were considering a double mastectomy. Two young people reported that they were booked in with a surgeon to discuss penile skin inversion vaginoplasty.

### 3.8. Rates of Persistence and Desistance

As of the November 2022–January 2023 follow-up (4–9 years post-presentation) of the 79 young people who had been referred to Psychological Medicine for diagnostic assessment for GD and potential gender-affirming medical interventions (with two lost to follow-up), 60 (60/77; 77.9%) were progressing on a GD (transgender) pathway, and 17 (17/77 (22.1%) were progressing on a diverse range of other pathways (see Figure 2).

Of the subgroup of 68 young people who had met DSM-5 criteria for a formal diagnosis of GD (with two lost to follow-up), 60 were progressing on a GD (transgender) pathway and six had desisted. For this subgroup of young people (GD subgroup), the persistence rate was 60/66, 90.9% (60/66), and the desistance rate was 9.1% (6/66).

One young person had desisted in the period in which they and their family were considering puberty suppression, three after having been started on puberty blockers, and three when on cross-sex hormones (see Table 2). In addition, one young person persisted in their transgender identify but decided to cease all medical interventions—that is, they desisted from the medical gender-affirming pathway (see Table 2). In the above statistics this young person is included in the participants who continued on the transgender pathway with social transition only.

### 3.9. Rates of Comorbid Mental Health Concerns on Follow-Up 

In our earlier study of this same cohort of 79 children seeking treatment for dysphoria regarding sex assigned at birth, we found, on initial diagnostic assessment (December 2013 to November 2018), that 70 of 79 (88.6%) received comorbid mental health diagnoses or displayed other indicators of psychological distress (see Table 3 in Kozlowska and colleagues 2021) [10]. 

On follow-up (November 2022–January 2023), current information about ongoing mental health concerns was available for 50 young people (see Table 3).

### 3.10. Educational/Occupational Outcomes

On follow-up (November 2022–January 2023), current information about educational/occupational outcomes was available for 50 young people. Of these, 24 (48.0%) were still at school; 10 (20.0%) were attending university; six (12.0%) were employed in blue- or white-collar jobs; two (4.0%) were undertaking an apprenticeship; two (4.0%) were enrolled in an assisted work program; and six (12.0%) were unemployed. Of those who were unemployed, all suffered from ongoing severe mental health concerns (including comorbid substance abuse in one case). This unemployment rate was only slightly higher than the Australian national average rate of 7.7% youth unemployment in November 2022 [73]. 

### 3.11. Sample Characteristics Viewed through the Lens of the Recent Swedish Guidelines

The recent Swedish guidelines, and the presentations meeting three key criteria—onset of gender-related distress in childhood; persistence over time; and high levels of distress with the commencement of puberty—have been termed “exceptional cases” (p. 3) [32]. Below we examine the make-up of our current cohort in light of these criteria.

In the current sample (n = 79), 41 (51.9%) young people had reported experiencing dysphoria about gender from toddlerhood or preschool age (toddler/preschool group); 22 (27.8%) from the early primary school years (primary school group); 12 (15.2%) as puberty approached or was in process in late primary school or early high school years (puberty-approaching group); and four (5.1%) when they were post-pubertal (rapid-onset adolescent group) (see Figure 2). With the exception of the small rapid-onset adolescent group, there was a slight preponderance of biological females across groups (see Figure 3).

Of the toddler/preschool and the primary school groups, 55 of 63 (87.3%) met diagnostic criteria for GD and fit the “exceptional case” category. Of this “exceptional case” group, 50 of 55 (90.9%) persisted on the GD pathway, 4 of 55 (7.3%) desisted, and 1 of 55 (1.8%) were lost to follow-up (see Figure 2).

Of the puberty-approaching group, 12 of 12 (100 %) met diagnostic criteria for GD. Because these young people’s dysphoria regarding sex assigned at birth had arisen when puberty had approached, they did not clearly fit into the exceptional-case category of the Swedish guidelines: the duration criterion was not met. Nor did this group fit into the rapid-onset presentations seen in adolescence. Of this group, 9 of 12 (75.0%) persisted on the GD pathway, 2 of 12 (16.3%) desisted, and 1 of 12 (8.3%) was lost to follow-up (see Figure 2).

Of the rapid-onset adolescent group (4/79; 5.1%), none met diagnostic criteria for GD (see Figure 2). All four young people were referred back to their individual therapists for further therapeutic work to address their distress, to support identity exploration, and to address comorbid mental health concerns.

## 4. Discussion

The Gender Service at the Sydney Children’s Hospitals Network was established in 2013 in response to an increase in presentations of children with gender-related distress seeking gender-affirming medical interventions. In this prospective case–cohort study, we followed the clinical, educational, and mental health outcomes of a cohort of 79 young people—13.25–23.75 years of age; 33 biological males and 46 biological females—who presented in the first five years (December 2013–November 2018) of the Gender Service. Following a triage process (see Figure 1), the diagnostic (biopsychosocial) assessment conducted by the Psychological Medicine team yielded a formal DSM-5 diagnosis of GD in 66 of the young people. Of the 13 who did not meet DSM-5 criteria, two obtained a GD diagnosis at a later time (at another service). This yielded 68 young people (68/79; 86.1%) with a formal diagnosis of GD (GD subgroup) potentially eligible for gender-affirming medical interventions, and 11 young people (11/79; 13.9%) who were not (non-GD subgroup). On follow-up in November 2022–January 2023, within the GD subgroup (n = 68, with two lost to follow-up), 60/66 (90.9%) had persisted on a GD (transgender) pathway, and 6/66 (9.1%) had desisted. Within the non-GD subgroup (n = 11), the young people had followed a range of developmental pathways (see Figure 2). In the cohort as a whole—the 79 young people who presented gender-related distress (with two lost to follow-up)—60 continued on a transgender pathway (overall persistence rate was 77.9% (60/77)), and 17 travelled an alternate pathway (overall desistance of gender-related distress was 22.1% [17/77]) (see Figure 2). Ongoing mental health concerns were reported by 44 of 50 (88.0%). Educational/occupations outcomes varied widely. In the discussion that follows we discuss some of the key themes emerging from this research.

The current study highlights the importance of comprehensive screening and the value of a comprehensive biopsychosocial (including family) assessment during a young person’s engagement with the Gender Service. The young people in the current sample were referred for a formal psychiatric diagnostic (biopsychosocial) assessment for GD because they were seeking gender-affirming medical interventions. The referral to Psychological Medicine was made after two sets of screening—phone triage and a screening medical assessment by a paediatrician. At that point in the young person’s journey, the biopsychosocial assessment is crucial in exploring the many different factors—predisposing, precipitating, and maintaining—that may have contributed to the young person’s distress and presentation [10]. In the 13 young people who did not meet the diagnostic criteria for GD at that time, the biopsychosocial assessment provided the young person (and their individual therapist and family) with a rich formulation to support ongoing identity exploration, pathway choices and treatment options to address high levels of distress, comorbid mental health issues, family issues, and school-related issues. In the 66 young people who did meet diagnostic criteria at that time—and who were eligible for the gender-affirming medical interventions that they were seeking—the formulation likewise identified a similar range of issues that needed to be addressed to ensure the best possible future outcomes (see Kozlowska et al. 2021) [10]. Taken together, the screening and psychosocial assessment enabled our Gender Service to work with young people and their families to distinguish between young people with gender-related distress for whom the gender-affirming medical was potentially appropriate and those for whom it was not. The broader research question concerning the benefits versus risks of gender-affirming medical treatment is one for ongoing reflection, research, and public policy (see the introductory summary above regarding the current situation in Finland, Sweden, the United Kingdom, Australia, and the United States) [31,32,33,34,48,49,50,51]. See also current conversations pertaining to these issues [52,67,68,69,70,71].

It is important to emphasize the importance of a neutral therapeutic stance that communicates the possibility of multiple developmental pathways and choices. The expectation is that a thorough, probing biopsychosocial assessment will yield a working formulation that evolves over time as more understanding is gained; the future developmental pathway and ultimate outcome are always left open. Along these lines, and in light of the relatively scarce and inconclusive evidence base regarding gender-affirming medical interventions, our multidisciplinary team made no a priori assumption that any particular young person was an appropriate candidate for gender-affirming medical interventions or that such interventions represented the only possible treatment pathway. A recurring message that we communicated to all young persons and their families—across clinical encounters—was that many pathways were potentially available to young persons experiencing gender distress.

A further message was that ongoing supportive psychotherapy functioned as a “safety net”: it created a space for helping the young persons to navigate their own development, including but not limited to their understanding of their own issues relating to gender and emerging sexuality. During the time frame of this study, all young people referred to Psychological Medicine for a diagnostic assessment were required to be engaged with a community-based therapist or mental health team who would support ongoing therapeutic work. The importance of ongoing exploratory psychotherapy (including family therapy) is emphasized in the new Finnish and Swedish guidelines, in the interim report from the Cass Review, and in the emerging literature.

Even so, a serious problem remains. Despite the most careful screening and biopsychosocial assessment, some young persons who seek gender-affirming medical interventions and become eligible for and receive these interventions will come to regret their earlier decisions and will choose to desist or detransition [57,59,60,63,64,65]. In the current sample (whole sample), at this point in the follow-up process, 17/76 (22.4%) had not pursued a transgender pathway, and the desistance rate in the subgroup who met diagnostic criteria for GD, was 6/66 (9.1%). For the group of young persons who persisted along the medical pathway, the consequences of receiving gender-affirming medical interventions—especially those that are irreversible—are likely to be life changing. If the choice of pathway was the right one for any particular young person, it may well support ongoing adaptation and wellbeing. If the choice of pathway was not the right one, it may seriously distort both the young person’s life choices and ongoing sense of wellbeing. By hypothesis, however, both sets of young persons—those whose medical interventions led to favourable outcomes and those whose interventions did not—had made choices that were to them, when made, medically and individually appropriate. It was only later that dissatisfaction with the earlier choices came into focus. Just how to address this challenge is currently an open question much in need of further research.

Given the above, in the Australian context, it is arguably ill-advised to “loosen up” the requirement for a psychiatric diagnostic (biopsychosocial) assessment for young people with gender-related distress seeking gender-affirming medical interventions. For example, under the *New South Wales LGBTIQ+ Health Strategy 2022–2027: Summary of Evidence*, all 79 participants in the current cohort would have had access to gender-affirming medical interventions—whether they met diagnostic criteria for GD or not—yielding, with time, a projected desistance rate of 22.1% (17/77). That is, under this framework, and using the present cohort as a reasonable sample, one could project that more than a fifth of the sample (17/77, or 22.1%, in our study) could have been exposed to inappropriate medical treatment, future regret, and potential harm.

A related theme concerns what critics of the gender-affirming medical pathway refer to as the self-fulfilling prophecy of initiating medical treatment with puberty blockers (p. 5) [74,75]—what Zucker has referred to as “treatment that is, in effect, iatrogenic” (p. 37) [76]. Additionally, Nahata and Quinn (2019) have suggested that young people on puberty suppression “may not have full developmental capacity due to lack of brain development (p. 759) [55]. Rather than serving, as intended, as a “pause button” that creates for the child and family a period for reflection and further consideration, these critics have argued that the use of puberty blockers sets children onto a medical treatment pathway that, for better or worse, they are unlikely to step away from—with major consequences for who they are and how they live. This reflection is an important one. In the current study, one child desisted whilst considering puberty blockers. Of the 49 children with a formal diagnosis of GD who opted for puberty suppression (stage 1), only three (6.1%) desisted whilst on puberty blockers, whereas 38 (77.6%) continued to treatment with cross-sex hormones (stage 2) and nine (18.34%) were waiting to turn 16 to be eligible for further assessment for treatment with cross-sex hormones. Thus, in the current sample, only a small number changed their minds during puberty suppression and desisted. This finding is consistent with other studies which show puberty blocker discontinuation rates and desistance from the gender affirming medical pathway of 1.6% [14]; 4% [77]; 7% [78] and 8% [79]. Whether our findings reflect a successful screening and assessment process that identifies “exceptional cases” (p. 3) [32], or whether they reflect the dynamic of the self-fulfilling prophecy dynamic alluded to above, cannot be determined at this time. Outcome data many years down the track are needed to clarify this complex question.

Puberty blockers have previously been considered a fully reversible treatment. Over the last decade, however, reports about adverse effects on bone density—leading in adulthood to increased risk of fracture and kyphosis—have caused some practitioners to reconsider this view [18,19]. In the current study adverse effects on bone health (7/49; 14.3%) and on cardiovascular health (via weight gain, 5/49; 10.2%) have emerged as two key areas of concern. Of interest was the finding that 4/7 participants with low bone density had low bone density prior to the initiation of puberty blockers (with further deterioration whilst on puberty blockers). Our clinical impression is that this subgroup of children included children who had tried to restrict their food intake as a means of delaying the onset of puberty and children whose lifestyles did not involve sufficient daily exercise. In this context, their bone health was already compromised at baseline, prior to the initiation of puberty suppression with GnRHa.

More broadly, many unknowns remain regarding the long-term effects of puberty blockers. An international, interdisciplinary team of experts has suggested the evaluation of neurodevelopmental effects as an urgent research priority [21]. Their consensus document suggests that three key domains—mental health, executive function/cognitive control, and social awareness/functioning—should be assessed at multiple time points using multiple comparison groups (untreated transgender youth matched on pubertal stage; cisgender youth matched on pubertal stage, and an independent sample from a large-scale youth development database). Another important domain that requires assessment is that of sexual function in adulthood, particularly in young people who have not completed endogenous puberty as a function of puberty suppression.

A decade ago, one argument pertaining to the use of gender-affirming medical interventions for GD was that these interventions alleviated the young person’s distress and led to improved psychological functioning. Early studies appeared to support this perspective [80,81,82,83,84], and this point of view is still widely held today [85]. More recently, however, a Finnish study by Kaltiala and colleagues suggested a more nuanced picture [30]. The study found that, following treatment with cross-sex hormones, the “need for treatment due to depression, anxiety and suicidality/self-harm was recorded less frequently... However, need for psychiatric treatment overall did not decrease from the level before and during the gender identity assessment to the real-life phase [cross-sex hormones]. New needs had also emerged about as frequently as need for treatment diminished” (pp. 5–6). Likewise, the current study suggests a more nuanced picture. In this cohort of young people with gender distress who presented for a diagnostic assessment for GD, treatment for ongoing mental health concerns was reported by 44/50 (88%) of young people, 4–9 years after initial presentation. The low rate of reported learning difficulties—which can be managed but which typically do not resolve per se—suggest that certain domains of problems may have been under-reported at follow-up. Furthermore, even if the 29 participants—for whom we did not have current information about mental health—were free of any mental health concerns, ongoing mental health concerns would still be found in more than half the sample (a hypothetical figure of 44/79, 55%). These findings are in line with the broader literature, where high levels of depression, anxiety, autism, and behavioural disorders are reported in young people with GD and gender distress more broadly [79,86,87,88,89,90,91,92,93,94].

And finally, it is important to note that of the six young people (6/66, or 9.1%) who desisted from the gender-affirming medical pathway, did so across three points in time: before starting puberty suppression; during puberty suppression, and during treatment for cross-sex hormones. The study period was insufficiently long to provide data about desistance from the gender-affirming medical pathway later in time. Nevertheless, recent stories from adult detransitioners suggest that a percentage of young people will make a choice to detransition during adulthood [57,58,59,60,61,62,63,64,65,66]. Because the risk of harm—irreversible physical changes due to cross-sex hormones or gender-affirming surgery—research pertaining this group of young people is particularly needed.

### Limitations 

This study presents a number of limitations. First, the study was a naturalistic follow-up study. It did not include a control group—sometimes known as ‘wait-and-see’ or ‘watchful waiting’ [76]—and it did not use a blinded, randomised approach. Notwithstanding this, the results raise many important questions for future research and for future reflection. Second, because gender-affirming cross-sex hormones had been prescribed to our former patients by endocrinologists and other providers external to our Gender Service (and never by us), the current study does not provide information about possible side effects experienced in relation to cross-sex hormones. We also do not have information about the number of participants who pursued fertility preservation. Third, we did not collect information about the perspectives of the therapists who worked with the young people and their families outside of our service. In this context, we do not know whether the therapists sought to hold a neutral therapeutic stance akin to our own or whether they held a different position or positions. It is possible that therapists’ own perspectives affected the patients’ decisions to choose to persist or desist, and also the quality of the intervention that the young person received for the treatment of comorbid mental health concerns. Fourth, because our follow-ups ended in November 2022–January 2023, we have information only about developmental pathway choices and outcomes—for example, the rates of persistence and desistance—from 4–9 years post-presentation. The youngest participant at follow-up was 13.25 years old, and the oldest 23.75 years old. We also do not know how many of the current sample will choose to have gender-affirming surgery (stage 3 treatment) once they reach adulthood. Fifth, an unanswered question in the paediatric literature is whether gender-affirming medical treatment improves or does not improve mental health outcomes and quality of life [48,49,50,51]. Because a substantial percentage of young people who had exited the service could not be contacted at this final follow-up time point, our data pertaining to current mental health concerns are limited to the young people that we were able to contact and to those who are still treated in the service. The same limitation applies to our data about educational/occupational outcomes. Sixth, in our telephone follow-up, we used open-ended questions pertaining to the young person’s current mental health. Hence, we can report only what the young people and their families reported spontaneously in response to these particular questions. Seventh, this study is unable to examine issues pertaining to any placebo effects that accompany medication use, in this case puberty blockers and cross sex hormones [95].

Despite these limitations, the study provides additional prospective data about young people’s pathway choices, side-effects associated with puberty blockers, and rates of persistence and desistance from a tertiary-care Gender Service located outside of the Netherlands (where most of the original data sets were collected and from where the bulk of current research originates) [14]. An important element of our study is that we were able to document the young people who had desisted from the gender-affirming medical pathway at various points in the process and the reasons for their choice to do so.

### 5. Conclusions

The data from this study show that when young people with gender distress present to health services seeking medical interventions, they end up following a diverse range of developmental pathways. The availability of gender-affirming medical interventions for the treatment of gender dysphoria is a recent one, evolving from the work of clinicians in the Netherlands [13,14,80,96]. Early studies have suggested that medical interventions were associated with positive outcomes [80,96]. This early body of work consequently served as the foundation for subsequent treatment guidelines and became established in medical systems via streamlined assessment processes and treatment pathways. The concept of medical affirmation was embedded in the broader culture by media and internet channels. Together, these processes gave young people with gender-related distress a clear message: “This is the best way to proceed,” and “The medical affirmation pathway will take away your gender dysphoria.” For many young people and their families, however, these messages favouring medical interventions, coupled with professionals’ affirmation of this pathway, potentially displaced their consideration of other options or other pathways [76].

The young people and families who presented to our service typically came to us with settled ideas concerning their prospective treatment pathways. In particular, based on what was known at the time, and given the severity of the young persons’ distress, they and their families considered medical treatment for gender dysphoria to be the single best option. In the last five years, however, the gender-affirming medical model has been questioned by both clinicians (who have highlighted the current lack of a solid evidence base [52,54,74,75,76]) and detransitioners (who have highlighted the potential for adverse outcomes [57,59,60,63,64,65]). The current evidence suggests the need for a much more nuanced and complex approach. As research data pertaining to long-term outcomes continues to accumulate, “the best way to proceed” is likely to be seen as ranging over a much more diverse range of treatment options and pathways, with each supported by a stronger evidence base than is currently available.

In conclusion, at the current point in time there is much to be done in the field of gender dysphoria. In the era of evidence-based medicine, the evidence-base pertaining to the gender-affirming medical pathway is sparce and, for the young people who may regret their choice of pathway at a future point in time, the risks for potential harm are significant. At the current point in time, key research priorities include: a better understanding of factors that underpin the recent increase in presentations in adolescent girls; the long-term effects of puberty blockers and cross-sex hormones on brain development and other physical parameters; long-term mental health outcomes; sexual function and fertility outcomes, and overall functional outcomes both for the subset of young people who remain content with the choice of medical gender affirmation and the subset who come to regret this choice of developmental pathway.

## Figures and Tables

**Figure 1 children-10-00314-f001:**
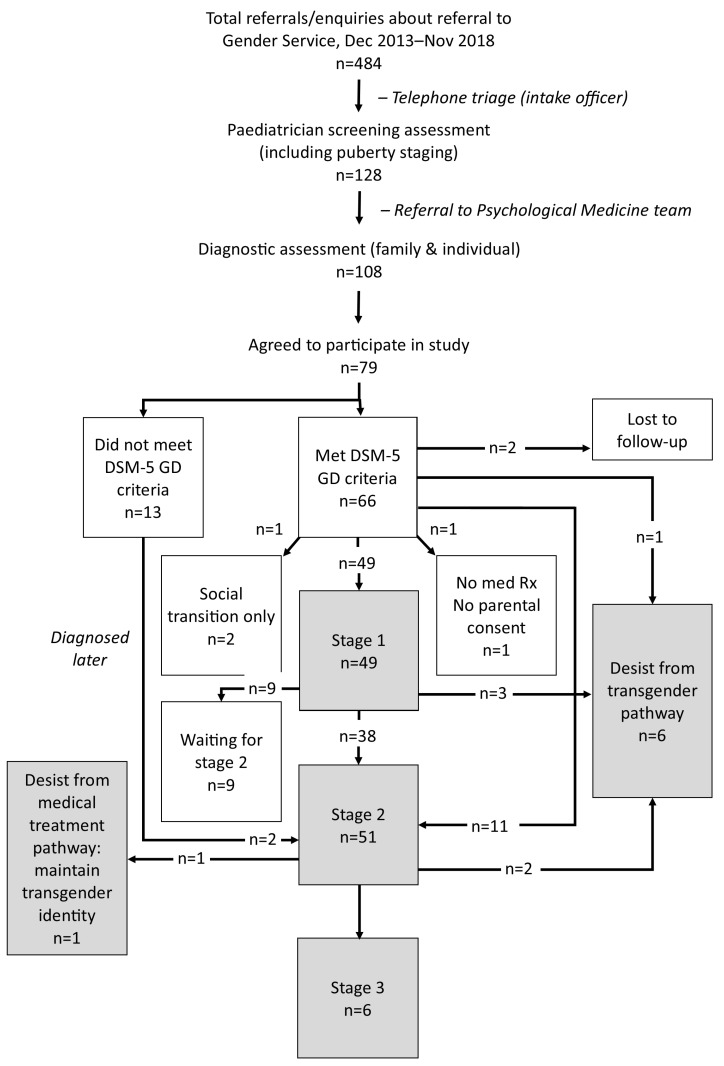
Flow chart summarising the triage, assessment, and treatment process.

**Figure 2 children-10-00314-f002:**
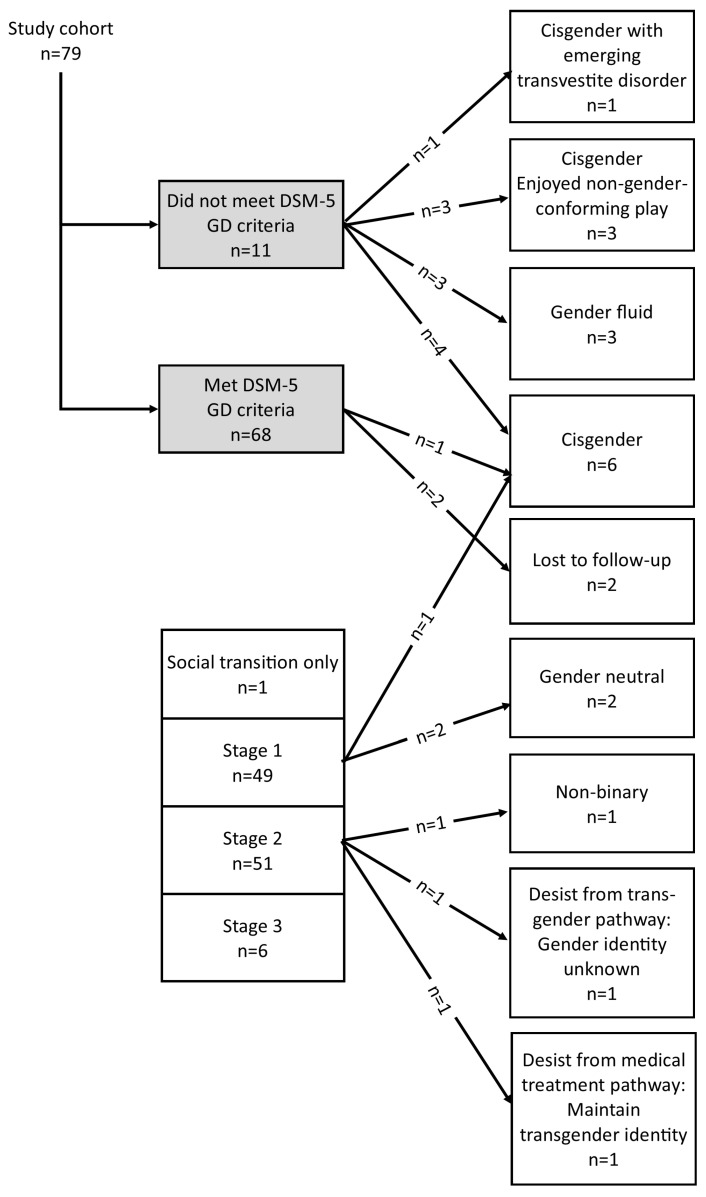
Developmental pathway choices of the young people.

**Figure 3 children-10-00314-f003:**
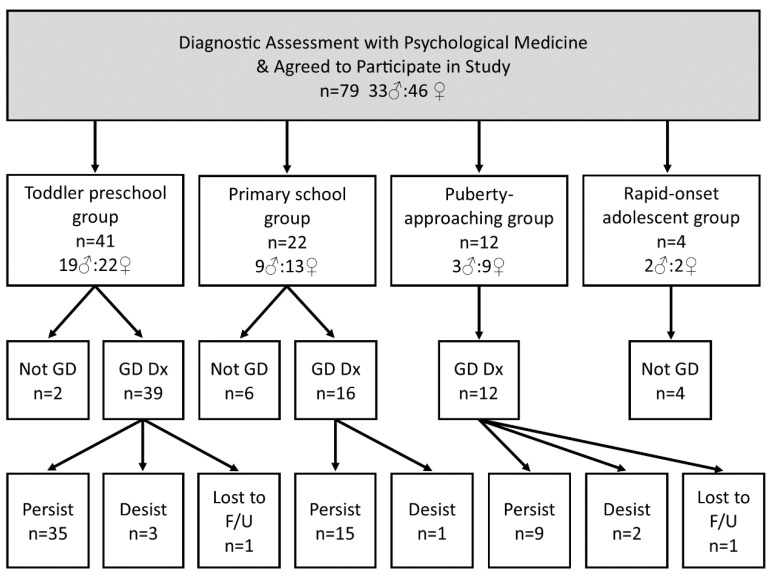
Flowchart showing assessment and treatment outcomes in the four different subgroups.

**Table 1 children-10-00314-t001:** Clinical details pertaining to the six participants with low bone density treated with puberty blockers.

**Subset of Young People with Low Bone Density Prior to Commencement of Puberty Blockers**			
Participant	Low baseline bone density (low prior to commencement of puberty blockers)	Further decrease in bone density following puberty suppression (low baseline bone density)	Decrease in bone density following puberty suppression (normal baseline bone density)
Case 1	Yes	Yes	--
Case 2	Yes	Yes	--
Case 4	Yes	Yes	--
Case 5	Yes	Yes (small deterioration only)	
**Subset of Young People with Normal Bone Density Prior to Commencement of Puberty Blockers**			
Case 3	No	--	Yes
Case 6	No	--	Yes
Case 7	No	--	Yes

**Table 2 children-10-00314-t002:** Clinical characteristics of the young people with GD who desisted from the gender-affirming medical pathway.

Biological Sex (♂/♀)	Age at Which the Medical Pathway Was Declined	Whilst Considering Puberty Suppression	During Puberty Suppression (Duration of Treatment)	During Cross-Sex Hormone Treatment (Duration of Treatment)	Stated Gender Identity at Time of Declining Medical Pathway
♀	12 years	√			Cisgender
♀	13 years		√ (1.83 years of PS)		Cisgender
♀	13 years		√ (1.08 years of PS)		Gender neutral
♀	15 years		√ (1.5 years of PS)		Gender neutral
♀	16 years			√ (2.33 years of PS and 4 months of CSH)	Transgender with social transition only *
♂	17 years			√ (4.75 years of PS and 8 months of CSH)	Non-binary
♀	18 years			√ (3.00 years of CSH)	Not known

√ = the tick symbol marks the time frame during which the young person decided to desist from the gender-affirming medical pathway. ♀ = female biological sex; ♂ = male biological sex; CSH = cross-sex hormones; PS = puberty suppression. * In the analyses this participant is included in the group of participants who continued on the transgender pathway with social transition only.

**Table 3 children-10-00314-t003:** Comorbid mental health diagnoses.

	Number (%) on Clinical Assessment in December 2013–November 2018 (Total n = 79)	Number (%) on Follow-Up (Reported Mental Health Concerns) in November/December 2022 (Total n = 50)
Comorbid MH diagnosis	70 (88.6%)	44 (88.0%)
No MH diagnosis	9 (11.4%)	7 (14.0%)
Anxiety	50 (63.3%)	22 (44%)
Depression	49 (62.0%)	25 (50%)
Any behavioural disorder (including ADHD, ODD)	28 (35.4%)	11 (22.0%)
Autism *	11 (13.9%)	15 (30%)
Learning difficulties **	8 (11.9%)	1 (2%)
Eating disorder	2 (2.5%)	2 (4%)
Psychosis	1 (1.3%)	0 (0%)
Substance abuse	--	1 (2%)
Intellectual disability	--	1 (2%)
Chronic fatigue syndrome	--	1 (2%)

ADHD = attention-deficit/hyperactivity disorder; MH = mental health; ODD = oppositional defiant disorder. * Since the initial presentation at the Gender Clinic, four new diagnoses of autism had been made—via formal autism assessments—mostly in follow-up to recommendations from the initial assessment. ** Learning difficulties can be managed but typically do not go away. In this context this low figure is reflective of under-reporting.

## Data Availability

The datasets presented in this article are not readily available because consent to place data in a public repository was not obtained from the children and families who participated in this study.

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
