# Peer review of "Developmental Pathway Choices of Young People Presenting to a Gender Service with Gender Distress: A Prospective Follow-Up Study"

_children, 2023, doi:10.3390/children10020314_

Round 1
Reviewer 1 Report
The authors present the developmental pathways of young people with gender identity distress (N=79). This is an area where data is much needed. The incidence of young people with gender identity issues has increased significantly, and we do not know much of the trajectories of outcomes of these patients.
The format is somewhat unusual. The introduction is long and there are no formal statistical significance testings of hypothesis. Rather, the authors first give a description of the current state of affairs in Australia, UK, US, Finland, and Sweden. They then give a detailed presentation of their cohort and present the trajectories of their 79 cases. They conclude that there are several possible paths that patients may take.
I believe, however, that this format is accurate given the current lack of data in this field. The data presented are, although scattered, important for the field for at least two reasons: First, there is a dearth of data from other countries than the Netherlands. Second, the cohort was included during the time when the incidence increased worldwide. Hence, it gives us information on this potentially ‘new’ group of mostly girls with gender identity issues.
My only comment is that I would appreciate if the authors tried to condense their text. It would facilitate reading and convey the message more clearly if themes were not unnecessarily repeated. For example, the second paragraph in the Discussion starts with that the current study underlines the importance of a comprehensive screening. The third paragraph starts with the same theme. This could be rephrased. If the authors invest some more work to present their findings as concisely and economically as possible, their important addition to the literature will be easier to digest.
Author Response
The authors present the developmental pathways of young people with gender identity distress (N=79). This is an area where data is much needed. The incidence of young people with gender identity issues has increased significantly, and we do not know much of the trajectories of outcomes of these patients.
The format is somewhat unusual. The introduction is long and there are no formal statistical significance testings of hypothesis. Rather, the authors first give a description of the current state of affairs in Australia, UK, US, Finland, and Sweden. They then give a detailed presentation of their cohort and present the trajectories of their 79 cases. They conclude that there are several possible paths that patients may take.
Reviewer 1 is quite right. There is very little data in this area and a lot of change on the policy public level system level across the globe. In this context we chose to inform the reader of the broader context in the introduction. We thought this broader context was very important. Our impression is that many people may not be aware of this broader discussion across different countries. We then provided the data pertaining to the pathway choices that the young people in our own study made.
Once again Reviewer 1 is correct, we took a naturalistic prospective follow-up approach. We did not try to make predictions or test hypotheses. The lack of current knowledge base and research data makes it difficult to set evidence-based hypotheses. In this context we thought that a naturalistic follow-up was appropriate. This naturalistic follow-up of our patients was also the research goal that we had set when the Gender Service opened. We have made this naturalistic approach more clear in the methodology section.
I believe, however, that this format is accurate given the current lack of data in this field. The data presented are, although scattered, important for the field for at least two reasons: First, there is a dearth of data from other countries than the Netherlands.
We have added a line at the end of the limitations section to underline the importance of having data from countries other than the Netherlands. We have also highlighted that we were interested in the choices of all young people. Those that chose to persist with the gender-affirming medical pathway and those that did not.
Second, the cohort was included during the time when the incidence increased worldwide. Hence, it gives us information on this potentially ‘new’ group of mostly girls with gender identity issues.
My only comment is that I would appreciate if the authors tried to condense their text. It would facilitate reading and convey the message more clearly if themes were not unnecessarily repeated. For example, the second paragraph in the Discussion starts with that the current study underlines the importance of a comprehensive screening. The third paragraph starts with the same theme. This could be rephrased. If the authors invest some more work to present their findings as concisely and economically as possible, their important addition to the literature will be easier to digest.
Thank you for this feedback. We have tried to condense some of the text in the discussion and get rid of unnecessary repetition. Please see redlining. We hope this works better.
We have retained the key elements of the paragraph about the neutral therapeutic stance. Our consumer representative thought this paragraph very important. We would like to respect the judgement of our consumer representative.
We also suggested that the reader looks at Figures as they digest the outcomes.
We have also added a missing arrow to Figure 1 which makes the data easier to read.
We also noted that MDPI had set our word version into their format but had repeated Figure 1 and omitted Figure 2. Please note that I have eliminated the second, incorrect occurrence of Figure 1 and substituted the correct figure, Figure 2.
We developed the flow chart figures to provide the data visually because the data are quite complex.
Reviewer 2 Report
Thank you for allowing me to review this manuscript. This manuscript entitled "Developmental pathway options of gender-distressed youth presenting to a gender service: a prospective follow-up study"
It is an interesting and highly relevant article today, although it has several limitations that make it susceptible to publication in this journal. These limitations are detailed below:
- The general quality of the article is very satisfactory. It contains a number of appropriate tables and figures, which collect the necessary information in a clear manner.
- As I indicated previously, it is a novel article and of great current interest. I would recommend that the authors justify the novelty and relevance of the study in more detail.
- I would recommend increasing the number of citations and their relevance in the discussion section.
- I would recommend that the lines of the future be highlighted in more detail in the conclusions.
I think the manuscript will capture the interest of the audience interested in this field.
good job
Author Response
Thank you for allowing me to review this manuscript. This manuscript entitled "Developmental pathway options of gender-distressed youth presenting to a gender service: a prospective follow-up study"
It is an interesting and highly relevant article today, although it has several limitations that make it susceptible to publication in this journal. These limitations are detailed below:
- The general quality of the article is very satisfactory. It contains a number of appropriate tables and figures, which collect the necessary information in a clear manner.
We are glad that Reviewer 2 thought that the Figures were helpful in digesting the results. We added a missing arrow figure to Figure 1.
We also noted that MDPI had set our word version into their format but had repeated Figure 1 and omitted Figure 2. Please note that I have eliminated the second, incorrect occurrence of Figure 1 and substituted the correct figure, Figure 2.
- As I indicated previously, it is a novel article and of great current interest. I would
recommend that the authors justify the novelty and relevance of the study in more detail.
We have added an additional paragraph to the discussion to take particular note of the young people who decided to desist from gender-affirming treatment. The results of our study—together with data from detransitioners—suggest that the decision to desist from gender-affirming treatment may happen at any point in the process. There has been little interest in this group of young people, and it is important that their pathway choices are also examined in research studies.
We have included a sentence about the novelty as a final sentence to the limitations to indicate that despite limitations, the study is important.
- I would recommend increasing the number of citations and their relevance in the discussion section.
We have increased citations throughout the article. This includes additional citations to articles about: comorbid mental illness (including autism); rates of desistance from puberty blockers; change in sex ratio over time; conversations between those who support gender-affirming medical treatment and those who don’t.
We also added a sentence to the limitations about our lack of information about puberty preservation. This is an important topic that we had omitted to mention. Likewise, we added the need for research pertaining to sexual function in adulthood.
We hope the addition of the references is satisfactory.
- I would recommend that the lines of the future be highlighted in more detail in the conclusions.
We have added an additional paragraph at the end highlighting the key areas of research that are needed.
I think the manuscript will capture the interest of the audience interested in this field.
good job
Round 2
Reviewer 2 Report
Gracias por tomar en cuenta las consideraciones y por realizar los cambios sugeridos.